# A Dataset for Analyzing Streaming Media Performance over HTTP/3 Browsers

**Sapna Chaudhary**
IIIT Delhi
sapnac@iiitd.ac.in

**Naval Kumar Shukla**
IIIT Delhi
naval19065@iiitd.ac.in

**Sandip Chakraborty**
IIT Kharagpur
sandipc@cse.iitkgp.ac.in

**Mukulika Maity**
IIIT Delhi
mukulika@iiitd.ac.in

## Abstract

HTTP/3 is a new application layer protocol supported by most browsers. It uses QUIC as an underlying transport protocol. QUIC provides multiple benefits, like faster connection establishment, reduced latency, and improved connection migration. Hence, popular browsers like Chrome/Chromium, Microsoft Edge, Apple Safari, and Mozilla Firefox have started supporting it. This paper presents an HTTP/3-supported browser dataset collection tool named H3B. It collects the application and network-level logs during YouTube streaming. We consider YouTube one of the most popular video streaming applications supporting QUIC. Using this tool, we collected a dataset of over 5936 YouTube sessions covering 5464 hours of streaming over 5 different geographical locations and 5 different bandwidth patterns. We believe our tool and as well as the dataset [1] could be used in multiple applications such as a better configuration of application/transport protocols based on the network conditions, intelligent integration of network and application, predicting YouTube's QoE, etc. We analyze the dataset and observe that during an HTTP/3 streaming, not all requests are served by HTTP/3. Instead, whenever the network condition is unfavorable, the browser chooses to *fallback*, and the application requests are transmitted using HTTP/2 over the old-standing transport protocol TCP. We observe that such switching of protocols impacts the performance of video streaming applications.

## 1 Introduction

HTTP/3 is the latest variant of the popular *Hypertext Transfer Protocol* (HTTP), which has recently been widely adopted by major Internet giants such as Google, Facebook, Cloudflare, Akamai, Apple, and many others. Unlike its predecessors, HTTP/3 uses QUIC [20, 14] as the underlying transport protocol. QUIC is expected to provide reduced latency and better application QoE (Quality of Experience) compared to TCP, the widely used transport protocol for the Internet, for the past three decades. The QUIC developers [23] and several other works [21, 37, 11, 39, 44] have shown the superiority of QUIC compared to TCP in terms of supporting better QoE with less stall for video streaming.

In this paper, we present a large-scale dataset containing network-application information for 5464 of YouTube streaming sessions over popular browsers supporting both HTTP/3 and the legacy HTTP/2. Notably, YouTube is one of the most popular streaming media services on the Internet today. To

---

[1]https://github.com/NKShukla/H3B (Access:December 19, 2023)

37th Conference on Neural Information Processing Systems (NeurIPS 2023) Track on Datasets and Benchmarks.

collect this large-scale dataset, we develop H3B, an emulation-based toolbox that captures the application performance statistics for YouTube video streaming coupled with the underlying network traffic traces. For this purpose, we explored a feature in the YouTube browser application, named *stats for nerds*, which shows the played video statistics in terms of the frames dropped, the current resolution, Internet connectivity in the device, and the YouTube playback buffer heath. However, it misses critical information like bitrate, variation in bitrate, and stalling/rebuffering used to compute the end user's QoE. Notably, the network bandwidth directly impacts YouTube *adaptive bitrate* (ABR) decisions [29], where the YouTube client dynamically decides the best playback bitrate depending on the underlying network conditions. Therefore, to analyze the application performance, one needs to see the network behavior as well to perform root cause analysis as and when application QoE suffers. Hence, we develop H3B that collects application and network layer logs simultaneously while streaming videos over YouTube. We collected the data over two web browsers – *Chrome/Chromium* and *Firefox*. The tool takes (1) video ID and (2) the network bandwidth pattern as the input and generates the application and network logs with annotated QoE information. H3B emulates that network behavior using a benchmark network emulation tool called *Mahimahi* [32].

We use H3B to launch a measurement campaign for YouTube across 5 different geographical locations and 5 different bandwidth patterns. Out of them, 2 are traces collected under mobility in WiFi and cellular networks. Specifically, we focus on the poor network bandwidth patterns as there is minimal data on video streaming applications' performance for poor networks [17, 42]. Further, several existing studies [28, 26, 2, 46] suggest that poor or fluctuating network conditions provide the opportunity to use intelligent learning-driven algorithms for optimizing the streaming media performance. Notably, the Internet speed is still deficient in significant parts of the globe, particularly the developing world [1, 8, 27, 34]. We have collected 5464 of streaming hours across 5936 streaming sessions of data through H3B. To benchmark the performance of YouTube streaming sessions over HTTP/3 browsers compared to the legacy HTTP/2, we have also collected the logs and traffic data over the HTTP/2 setup under similar network configurations.

While analyzing the impact of the HTTP/3 protocol on YouTube performance, we have a few interesting observations from the dataset collected above. We observe that during HTTP/3 streaming, even though it is assumed that it will use QUIC underneath, we observe that often the data is sent over TCP. Such a phenomenon is called *fallback* [23, 18] where QUIC uses the help of TCP on a network path where UDP (QUIC) is blocked. However, there was no such blocking in our setup, yet the browser chooses to fall back to TCP whenever the application suffers over QUIC. We next compute the QoE obtained by both HTTP/3 and legacy HTTP/2 streaming and perform hypothesis testing on QoE obtained for both. We observe that QoE obtained over HTTP/3 streaming is not a winner, contradicting the observations made by prior work and QUIC developers themselves [23, 35, 3, 40, 36, 19, 25]. We expect that the browser implementation of the legacy support for HTTP/3 might be a cause for the same. To further validate whether indeed this protocol fallback was the cause for poor application QoE, we modified the *Chromium* browser source code to stop such protocol switching forcibly. Then, using our tool H3B, we further launch a measurement campaign over this modified browser. We observe that forcibly stopping the protocol switching often improves the application QoE compared to the original one.

To the best of our knowledge, this is the first large-scale YouTube streaming dataset over an HTTP/3 browser. We have also collected the benchmarked HTTP/2 traffic (legacy traffic by disabling QUIC on the browser) for all the equivalent scenarios, which can be used to explore the pros and cons of various network configurations and protocol design choices over an HTTP/3 browser. Overall, our dataset has the following attributes:

- **Multi-bandwidths**: The dataset contains the application and network logs under different bandwidth patterns, i.e., high, low, very low, and under mobility (over WiFi and cellular network). This can be used to study how different network bandwidth patterns impact YouTube's QoE.

- **Multi-locations**: The dataset was collected for 5 different geographical locations, i.e., Delhi, Bangalore, New York, Germany, and Singapore. This can be used to study whether location plays any role in YouTube's QoE.

- **Multi-protocol**: The dataset contains both HTTP/3 and legacy HTTP/2. This can be used to analyze benefits/issues with HTTP/3.

- **Multi-video**: The dataset was collected for 46 videos of different genres: News, Entertainment, and Education. This can be used to study the impact of video type on YouTube QoE.

- **Time interval**: This data collection happened for 18 months. This allows a time variance study of the QoE during the course of browser updates.

Such a dataset can be used to develop intelligent models for network-application integration over an HTTP/3 browser while considering the backward compatibility with HTTP/2. In general, such a dataset can be used for the following problems:

- *Dynamic tuning of protocol design choices/configurations based on the network conditions*: This will allow learning the network environment to tune the parameters of, say, transport protocol congestion control, ABR (Adaptive BitRate) streaming parameters, etc.

- *Intelligent network-application integration for better application QoE*: Simultaneous logs of the network and application layer will facilitate designing network-aware applications.

- *Prediction & Optimization of YouTube QoE*: Further, given the diversity of our dataset, the same can be used to develop a prediction model for predicting QoE. Further, our poor network dataset provides an opportunity to use intelligent learning-driven algorithms for optimizing QoE.

## 2   Related Work

We divide the related work into three dataset categories: YouTube, DASH, and QoE datasets. **YouTube Dataset:** Gutterman *et al.* [17] works on the prediction of video QoE such as buffer state, quality of a video, stalling to be experienced. They collected data for 425 video sessions over YouTube for WiFi and LTE networks under static and mobility scenarios. Karagkioules *et al.* [22] collected a dataset of around 374 hours of YouTube videos on a mobile device using their tool *Wrapper-app*. The authors have collected the data from the network and application layer and extracted the application logs via stats for nerds and DNS queries. They have experimented at different bandwidth levels: 500kbit/s, 1024kbit/s, 3000kbit/s, and 100kbit/s. Wassermann *et al.* [42] used an app called *YoMoApp* [41] for collecting datasets of YouTube streaming over a cellular network. They collected a dataset of over 360 different mobile users, over 70 cellular network operators, and a total of 3000 video sessions. The author replicated the design and functionalities of the YouTube application for data collection. Though the prior work has focused on data collection of YouTube; there needs to be a specific focus on the poor or variable network bandwidth observed in developing countries. Further, we found that the existing dataset misses critical QoE information such as [22] misses stalling information and [42] uses the resolution not the birates which provides a more detailed assessment of quality. Moreover, some dataset does not use the production endpoint of YouTube [41].

**DASH Dataset:** Taraghi *et al.* [38] released a dataset containing different video codecs and bitrates with a maximum resolution of 8K. Feuvre *et al.* [24] release a dataset of HEVC, which was ranging from HD to UHD bitrates. Such MPEG-DASH packaged content dataset allows (1) efficient usage of these codecs, as not all devices support all the available codecs, (2) experimenting with different DASH adaptation techniques which support several codecs. DASH dataset is complementary to our dataset as in the comparison between HTTP/3 and legacy, the DASH algo and the codec used for the streamed video remain the same. Our dataset can boost DASH algorithms during poor network and protocol switchings.

**QoE dataset:** These datasets can be used for improving the rate adaptation of DASH algorithms and are collected using two traditional video quality assessment techniques. (1) Subjective assessment: Such assessment typically uses a MOS score that represents video quality perceived by the end user. The prior work typically creates a dataset of high-quality videos along with their distorted videos where they incorporate quality change/degradation and/or stalling events. Chen *et al.* [10] designed a model to predict the time-varying subjective quality (TVSQ value). Duanmu *et al.* [13] proposed a streaming QoE index, which accounts for the instantaneous degradation of quality and the initial rebuffering and stalling perceived by the end user. Bampis *et al.* [6] have created a database of Netflix videos and prepared distorted videos by imposing different playout patterns such as imposing different compression rates, rebuffering rates. This dataset simulates the real network by using different bandwidth patterns. They compute the MOS score from the subjective evaluation with respect to the frame index for all the distorted videos. (2) Objective assessment: such assessment computes quality scores such as PSNR and SSIM. Bampis *et al.* [5] have created a database of 420 distorted videos. They compute continuous and retrospective prediction scores such as MOS, PSNR, and SSIM. The dataset also includes the number of rebuffering events and different playback bitrates.

QoE datasets differ from ours (1) we have not *artificially* created distorted video dataset; rather, we use realistic bandwidth patterns that causes quality drop and stalling instances while streaming videos over YouTube. Different QoE datasets either have no stalling [10] or fixed stalling events at fixed durations [13], or fixed stalling patterns [15] (2) Length of a video is at max 300 sec, whereas each of our video duration is about 3000 sec (3) None of the datasets have network logs in addition to application logs. We have time-synchronized application and network logs that can be used to better characterize the impact of the network on application QoE (4) All datasets are of HAS (HTTP Adaptive Streaming) with only one version of HTTP, we provide with two HTTP protocols and two web browsers, different locations (5) The datasets contain QoE information in an aggregated fashion that lacks temporal patterns such as [5] provides only one rebuffering duration for the entire video.

## 3 H3B Tool

In this section, we present the design of our tool H3B. The tool takes video id and network bandwidth pattern as input and provides application layer logs regarding application QoE. Network layer logs in terms of packet exchanges with protocol as output. Details follow:

### 3.1 Input to H3B

**YouTube video selection:** We first create a list of 46 YouTube videos, each lasting 40 minutes to 1 hour as shown in Table 1. The genres of videos are News, Entertainment, Education, Indian talk shows, Comedy, Stanford online lectures, and British TV series. The minimum and maximum video quality of all the videos were 144p and 1080p. We use the YouTube developer's API to fetch necessary information about a particular video using its unique identifier. We made HTTP GET request [2] with the video's unique identifier. It is important to note that this endpoint is no longer available, maybe due to changes in YouTube's policies. We obtain a mapping between itag, bitrate, and the video quality corresponding to a particular video using the YouTube developers API. Table 2 shows the mapping of itag to the corresponding bitrate and the quality label of a video. Multiple quality labels include 144p, 240p, 360p, 480p, 720p, and 1080p. YouTube supports both *constant bitrate* (CBR) and *variable bitrate* (VBR) encoding; thus, a same quality label can have multiple bitrates and hence multiple *itag*s. For example, Table 2 shows three different bitrates for each quality label and corresponding *itag*s.

Table 1: Details of the selected videos

| | |
|---|---|
| **Number of Videos** | *46* |
| **Video duration** | *40 minutes - 1 hour* |
| **Types of Videos** | *News, Entertainment and Education videos* |
| **Minimum Video Quality** | *144p* |
| **Maximum Video Quality** | *1080p* |

Table 2: Video-Info Table of two sample videos

| Video ID —> | -SI0HKTfHN4 | | | | | | | | | | | | | | | | |
|---|---|---|---|---|---|---|---|---|---|---|---|---|---|---|---|---|---|
| Itag | 137 | 22 | 135 | 134 | 133 | 160 | 18 | 136 | 242 | 136 | 398 | 244 | 397 | 243 | 396 | 278 | 394 |
| Bitrate | 4466585 | 743210 | 930845 | 654628 | 301679 | 121826 | 576066 | 2029085 | 244471 | 2029085 | 1497188 | 849429 | 737405 | 492803 | 399554 | 112110 | 94436 |
| Quality | 1080p | 720p | 480p | 360p | 240p | 144p | 360p | 720p | 240p | 720p | 720p | 480p | 480p | 360p | 360p | 144p | 144p |
| Video ID —> | 4QcHHal-pt4 | | | | | | | | | | | | | | | | |
| Itag | 248 | 247 | 243 | 242 | 278 | 18 | 22 | 137 | 136 | 135 | 397 | 134 | 396 | 133 | 395 | 278 | |
| Bitrate | 2680531 | 1503593 | 756822 | 412207 | 224867 | 98699 | 512759 | 649740 | 4466452 | 1804135 | 1019629 | 706715 | 603781 | 387958 | 303545 | 185179 | 98699 |
| Quality | 1080p | 720p | 480p | 360p | 240p | 144p | 360p | 720p | 1080p | 720p | 480p | 480p | 360p | 360p | 240p | 240p | 144p |

**Network emulation:** The purpose of such an H3B tool is to replay a given network behavior and stream YouTube videos on that network. While it is not possible to replay bandwidth patterns over an "in-the-wild" setup, a large number of recent studies [28, 12, 31, 45, 4, 9] have relied on benchmark network emulator frameworks like *Mahimahi* [33] to analyze the protocol performance in a realistic setup. Accordingly, to emulate a specific network bandwidth over the test setup, we use *Mahimahi* Link Shell (`mm-link`) emulation, which emulates network links using user-specified packet delivery trace files. Mahimahi maintains two queues – one for the uplink traffic and the second for the

---

[2]$https : //www.youtube.com/get\_video\_info?video\_id = \%s\&el = embedded\&ps = default\&eurl = \&gl = US\&hl = en\&html5 = 1\&c = TVHTML5\&cver = 6.20180913"\% - SI0HKTfHN4$

downlink traffic. Whenever packets arrive from the Mahimahi's `mm-link` or Internet, it is placed directly into one of two packet queues depending upon whether it is uplink or downlink. Then it releases the packets based on the input packet-delivery trace file. So, each line in a packet-delivery trace file represents the time at which the packet of size MTU can be delivered. Also, `mm-link` wraps to the beginning of the input packet-delivery trace file on reaching its end. We write a Python script to generate such packet-delivery trace file to support the corresponding network bandwidth. Other than emulating bandwidth patterns, we support emulating any natural network packet trace collected using a packet capture tool such as *Wireshark* or `tcpdump`. We converted the packet traces to packet-delivery trace files using the mechanism used in [28].

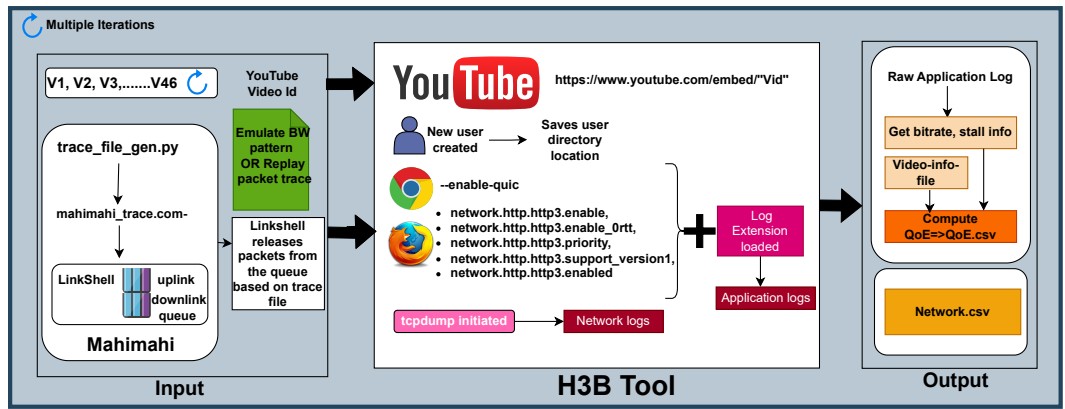

Figure 1: H3B architecture

### 3.2 H3B tool

H3B first creates a new user profile and then saves the location of the user data directory, where application logs are to be stored. It allows enabling and disabling QUIC while streaming the videos. In Chrome/Chromium, the `-enable-quic` flag is employed, while in Firefox, a set of preferences including `network.http.http3.enable`, `network.http.http3.enable_0rtt`, `network.http.http3.priority`, `network.http.http3.support_version1`, and `network.http.http3.enabled` are adjusted accordingly. Notably, when we disable QUIC, the browser setup uses the legacy HTTP/2 instead of HTTP/3. In order to embed the YouTube video inside the browser, we created our I-Frame and appended the YouTube video id at the end of "`https://www.youtube.com/embed`", which is called an I-Frame source URL [43]. The autoplay was also set inside the I-frame to play the video once the player was loaded automatically. We collect the application logs by creating a *log extension* and then integrate it into the browser. Loading this log extension was easier in Chromium, but in the case of Firefox, it was a difficult task. Therefore, for Firefox, we use the command-line tool web-ext to run this extension with the `-verbose` flag to print the logs in the terminal. We have used console.log API inside the log extension to collect the logs. It logs all the HTTP requests and responses between the client and the server. Inside the logs, we observe two types of requests (1) the video playback request, which contains the video segment information, and (2) the QoE request. In addition to the application level logs, we collect the network level logs. We use the packet capture tool `tcpdump` to collect the network packet captures (pcap). On the completion of video streaming, the application and network logs are stored in the local file directory, terminating the `tcpdump` process along with the user profile.

### 3.3 Output of H3B

H3B generates an application log and the corresponding network log at its output.

```
"52": {
    "type": "videoplayback",
    "request_ts": 6.583864990234375,
    "complete_ts": 13.9705791015625,
    "total_time": 7.386714111328125,
    "total_bytes": 174508,
    "complete_range": [
        0,
        174508
    ],
    "complete_itag": 397,
    "complete_rbuf_sec": 0.0,
    "complete_rbuf": 0,
    "complete_rn": 1,
    "complete_clen": 150413396,
    "complete_dur": 2624.64,
    "kbytes/second": 23070.876465714333
},

    "58": {
        "type": "streamingstats",
        "request_ts": 10.362389892578125,
        "itag": 397,
        "buffer_health": "10.027:0.00",
        "cmt": "10.027:0.000",
        "bwe": "10.027:130000",
        "vps": "10.027:B"
    },
```

**Application log structure:** The application logs provide two types of information. (1) *Video-related information while streaming a YouTube video*: The timestamp of the requested segment, total bytes of the segment, the itag value (tells the audio and video quality), the duration of the requested segment, and the protocol (TCP or QUIC) it uses for the segment request. (2) *Statistics about video streaming*: Information like the amount of video data that has been rendered and has been played, the quality of the segment stored in the buffer, buffer health (tell at any time $t$ how much amount of video is buffered in the buffer), and the playback duration in terms of how much video has been played. The sample application log description is shown above, and the details of important parameters in Table 3.

Table 3: Application log description

| Application log parameter | Description |
|---|---|
| request_ts | the request timestamp of the requested segment |
| complete_ts | the complete timestamp of the requested segment |
| total_time | the total timestamp from segment request to segment request gets complete |
| total_bytes | the total bytes of the segment |
| range | bytes of the segment data to be downloaded |
| itag | the requested video segment quality |
| rbuf | the receiver buffer in seconds |
| clen | the maximum possible length of the requested segment |
| dur | duration of the downloaded segment |
| buffer_health | at real-time t for how much duration the video has been buffered |
| cmt | at real-time t how much duration of the video has been played |

We convert the raw application log into a JSON file by only extracting the required features. We compute QoE using the application logs and video_info file shown in Table 2 that provides *itag* to bitrate mapping. Thus, we obtain QoE.csv that contains the average bitrate, average bitrate variation, average stall, and QoE. We use the formula-based QoE (Quality of Experience) metric used in Pensieve [28]; $QoE = Avg.\ Bitrate − Avg.\ Bitrate\ Variation − 4.3 × Avg.\ Stall$

Various QoE formulas used in previous literature are mentioned in [7]. The prior work either provides more preference to bitrate, bitrate variation, or to rebuffering. We calculate QoE at the chunk level not providing more preference to any one factor. [7] discusses the influencing factors such as quality switching frequency, quality switching magnitude, quality switching direction, duration of rebuffering, frequency of rebuffering, bitrate, initial delay, and user engagement. Our QoE metric considers most of them, including initial delay which is nothing but the initial stall duration. Since we are computing the average bitrate, variation, and stall the impact of factors like quality switching direction, and frequency of rebuffering also gets included.

**Network log structure:** The network logs are packet captures (pcap) obtained using tcpdump. Packet captures contain detailed information about a packet such as timestamp, source, and destination IP address, port number, application protocol (HTTP), transport protocol (TCP or QUIC or UDP), fields specific to TCP (SYN, FIN, RST, etc.), packet length, sequence number, acknowledgment

number (for TCP, for QUIC one has to decrypt the packet header), packet type (TCP data/ACK, QUIC handshake/initial/payload, etc.), whether retransmitted, etc. Since we are interested in correlating the number of bytes transferred over two transport protocols TCP and QUIC with application layer QoE, we extract five fields and convert them into a csv format. The structure of the network.csv is shown below:

```
Timestamp,  Source IP,  Destination IP, protocol, length
03:04:42, 192.168.29.48, 142.251.12.188, QUIC,     559
```

## 4    Dataset Description

We launched a measurement campaign using H3B over 5 different geographical locations. The locations are: Delhi, Bangalore, New York, Germany, and Singapore. Such a measurement was conducted using two different web browsers Chrome/Chromium and Firefox. We stream once over HTTP/3 and once over legacy HTTP/2. One of the testbeds is set up inside our campus premises, and the rest are set up using Digital Ocean machines. We emulate 5 different bandwidth patterns namely Dynamic High (DH): a good bandwidth, Dynamic Low (DL): a poor bandwidth and Dynamic Very Low (DVL): a very poor bandwidth, Real: real packet captures under mobility over WiFi and over cellular network.

**Bandwidth patterns:** To emulate DH, DL, and DVL bandwidth patterns, we created user-specified packet delivery trace files. Table 4 shows the bandwidth patterns where each pattern has the starting bandwidth, last bandwidth, and the Jump required to move from the starting bandwidth to the last bandwidth. After each jump, that bandwidth stays at that bandwidth for the Jump duration. This pattern from start to last bandwidth and then back from last to start bandwidth repeats in a cyclic fashion. Note that to emulate these bandwidth patterns, we use the Mahimahi network emulation tool. We also replay real packet captures. The real packet captures are of two categories, over WiFi and cellular network. Since we focus on a poor network, we utilize 105 mobility traces collected while watching YouTube videos over WiFi [16]. For cellular networks, we collected "in-the-wild" traces from 10 Android smartphone users watching YouTube videos of their choice using the cellular network. The volunteers are from 4 developing countries namely Kenya, Saudi Arabia, Pakistan, and India. The data was collected from 14 cities. Note that we focus on low and middle-income developing countries for collecting the data. The volunteers were instructed to collect network traces (pcap) using a *pcapDroid* application on their smartphones while watching the YouTube videos. The data was collected while the volunteers were traveling to/from their workspace to home. The phones used by volunteers had android versions 9-12

Table 4: Bandwidth Patterns

| Bandwidth Pattern | Starting Bandwidth | Last Bandwidth | Jump (Kbps) | Jump Duration |
|---|---|---|---|---|
| *Dynamic High (DH)* | 1152Kbps | 896Kbps | -256 | 240 sec |
| *Dynamic Low (DL)* | 640Kbps | 128Kbps | -256 | 240 sec |
| *Dynamic Very Low (DVL)* | 64Kbps | 256Kbps | +64 | 60 sec |
| *Real* | Mobility Traces from [16] and "in-the-wild" volunteer traces | | | |

**Dataset structure:** As part of the campaign, we obtain application and network level dataset of 5936 sessions (Firefox: 100 over HTTP/3 + 100 over HTTP/2, Chromium: 1864 over HTTP/3 + 1864 over HTTP/2, Original browser: 1004 over HTTP/3 + modified browser: 1004 over HTTP/3) for a total duration of 5464 hours. Fig. 2 (a) shows the hierarchy of the dataset collected. $FS1, FS2..., and FS100$ are the streaming session pairs over the Firefox browser. $S1, S2, S3...., and S1864$ are the streaming session pairs over Chromium browser. Each streaming session pair consists of streaming once over HTTP/3 and once over HTTP/2. Fig 2(b) shows the hierarchy of the dataset collected for original and modified chromium browser over HTTP/3. $S1865, S1866, ...., and S2868$ are the streaming session pairs over the original & modified chromium browser. Table 5 shows the duration of the collected dataset over different browsers, locations, and network conditions.

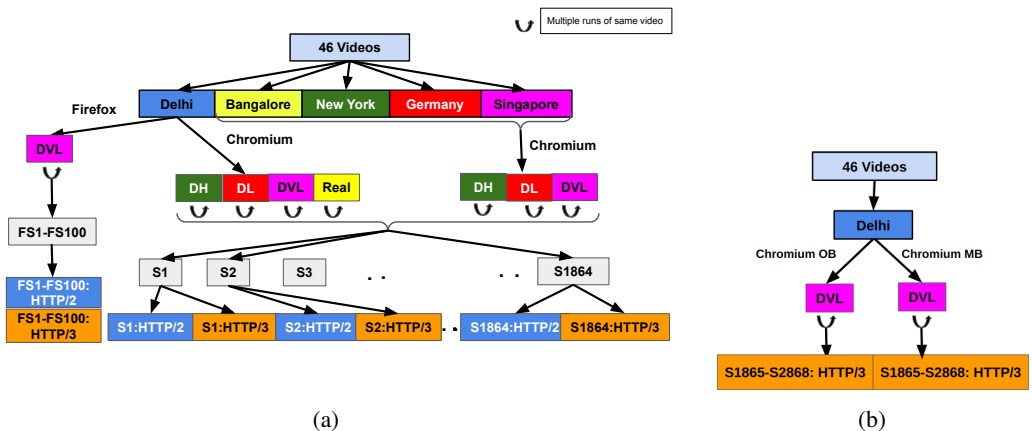

(a)
(b)

Figure 2: Dataset Hierarchy of (a) 1864 video session pairs over chromium and 100 video session pairs over Firefox browser on DH, DL, DVL and Real bandwidth patterns across Delhi, Bangalore, Singapore, Germany, and New York. (b) 1004 video session pairs over modified and original HTTP/3 enabled chromium-browser in Delhi on DVL bandwidth pattern.

Table 5: Dataset Details

| Configuration | Duration | Configuration | Duration |
|---|---|---|---|
| Chromium Delhi duration | 3941 hours | Firefox DVL duration | 142 hours |
| Chromium Bangalore duration | 490 hours | Chromium Total DH duration | 278 hours |
| Chromium Singapore duration | 253 hours | Chromium Total DL duration | 1712 hours |
| Chromium Germany duration | 459 hours | Chromium Total DVL duration | 157 hours |
| Chromium New York duration | 218 hours | Chromium Total Real duration | 894 hours |
| Chromium Total OB duration | 738 hours | Chromium Total MB duration | 715 hours |

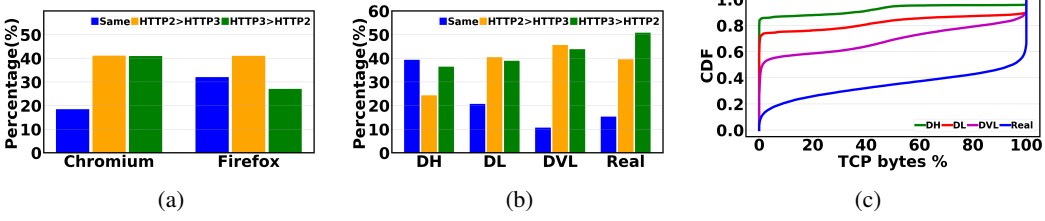

(a)
(b)
(c)

Figure 3: Hypothesis testing results over 1964 streaming session pairs on QoE over (a) Chromium and Firefox Browser, (b) different bandwidth patterns, and (c) TCP Fallback over Chrome/Chromium: Percentage of bytes transferred over TCP across all HTTP/3 streaming sessions for various bandwidth patterns. DH: Dynamic High, DL: Dynamic Low, DVL: Dynamic Very Low, and Real.

## 5 Dataset Analysis

We now use our dataset for analyzing the performance of HTTP/3. For the same, we compare the performance of HTTP/3 with legacy HTTP/2.

**QoE of HTTP/3 vs legacy HTTP/2:** To compare the QoE obtained over HTTP/3 and HTTP/2 statistically, we perform hypothesis testing over all 1964 streaming session pairs. We find out in what percentage of sessions HTTP/3 provides (a) better, (b) statistically similar, and (c) worse QoE compared to HTTP/2. Case (b) is considered the null hypothesis and vice versa as an alternative hypothesis. If the null hypothesis gets rejected, then we perform the two-tail test [30] to check whether the HTTP/3-enabled browser performs better. Fig. 3(a) shows the hypothesis-testing results on computed QoE values over Chromium and Firefox browsers. In the case of Chromium, we observe that for 81.5% of the cases, the two browsers behave differently. For 41% of the cases, we found the legacy HTTP/2 browser outperforms HTTP/3 one. In the case of Firefox, we have a similar observation that HTTP/3 is not always a winner. When we look into the network logs for these 41%

cases where HTTP/3 underperforms legacy HTTP/2: there were several instances of QUIC to TCP and TCP to QUIC protocol switching. From Fig. 3(b), we observe that for the scenarios under DL and DVL, HTTP/2 performs better than HTTP/3 for more than $40\%$ of the video session pairs. Fig. 3(c) shows the CDF graph of the percentage of TCP bytes experienced at various bandwidth patterns. We observe that for *DH*, there is the almost negligible presence of TCP traffic for $80\%$ times. Indeed, it is expected that QUIC does not face many connection failures for high bandwidth. For *DL*, *DVL* and *Real*, the presence of TCP traffic is $32\%$, $52\%$, and $98\%$ respectively for $70\%$ times. Fig. 4(a) indicates that HTTP/3 outperformed legacy for more than $40\%$ cases in Bangalore and Germany. However, across all five geographical regions, there are more than $30\%$ cases when legacy yielded better application QoE compared to HTTP/3. Again, correlating this TCP fallback in Fig. 4(b), we observe that Delhi has more instances of TCP traffic within HTTP/3 enabled streaming, compared to other locations; Fig 4(a) shows that HTTP/2 provides better QoE for these two cities (Delhi and Singapore) compared to HTTP/3. Thus, we observe that protocol switching impacts video streaming performance.

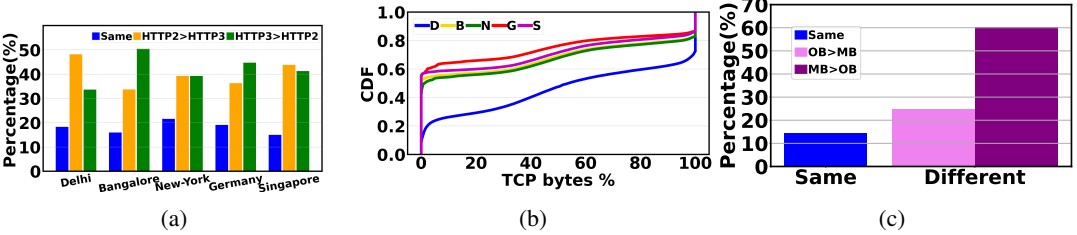

(a)                       (b)                       (c)

Figure 4: (a) Hypothesis testing results over different geographical locations. (b) TCP Fallback over Chrome/Chromium across locations. D: Delhi, B: Bangalore, N: New York, G: Germany and S: Singapore and (c) Hypothesis testing results over 1004 streaming session pairs on QoE of Original Chromium Browser (OB) and Modified Chromium Browser (MB)

**QoE of HTTP/3 supported original vs modified browser:** We observe that such a protocol switching occurs due to faulty implementation of fallback at the browser. Hence, an HTTP/3 browser tends to fall back to TCP at a low-bandwidth network. We, therefore, modify the Chromium source code by disabling the fallback completely. We then launch another campaign using H3B. We obtain 1004 YouTube session pairs over original and modified browser. We then perform hypothesis testing and observe that HTTP/3 modified browser outperforms original browser for $60\%$ of cases. Thus, we conclude that fallback to TCP hinders achieving the benefits of HTTP/3.

# 6 Discussion

We now discuss the limitations of our dataset, ethical considerations, and how the dataset can be utilized for future research.

## 6.1 Data Limitations

**(1) More diversity in networking conditions:** Though our dataset comprises various bandwidth patterns with a specific focus on poor bandwidth, there can be more diversity by including high bandwidth network types and other poor network bandwidth types. We allow replaying real packet traces; collected under mobility for both WiFi and cellular. This can be extended further with diverse traces collected under various mobility/poor bandwidth scenarios. **(2) More diversity in locations:** We have collected data for 5 locations. There can be much more diversity in locations focusing on locations from developing countries. **(3) Diversity in platform:** Our data collection was performed from a desktop platform; it can also be extended to include mobile platforms.

## 6.2 Ethical Considerations

This paper does not directly interact with human subjects or use any network services beyond their usage restrictions. All the network services used in this work (Digital Ocean machines) have been paid as per the usage.

### 6.3 Research Topics

**Dynamic tuning of protocol configurations based on learning the network environment:** Given the diversity of our dataset in terms of different network conditions, this can help better configure the protocol hyper-parameters used for video streaming. For example, to provide better application QoE, one can analyze the network conditions to tune hyperparameters like deciding the optimal transport protocol i.e., QUIC or TCP, tuning the transport protocol's congestion control or tuning the ABR (Adaptive Bitrate Streaming) parameters, and so on.

**Intelligent network-application integration:** We believe our dataset can allow intelligent network and application integration. To provide better application QoE, the applications should adapt themselves to network conditions. Since our dataset has both types of logs, this can enable better design of applications that will possibly look for *signatures* in the network behavior and adapt accordingly.

**Predicting and optimizing YouTube's QoE:** Our dataset provides the QoE of YouTube applications under different network bandwidth patterns. Utilizing such data, one can develop a learning model to predict YouTube's QoE. Further, our dataset for poor or variable networks provides an opportunity to use intelligent learning techniques for optimizing YouTube QoE.

**Predicting QoE of HTTP/3 based video streaming application:** Our dataset contains the network packet exchanges and the corresponding QoE. We believe our dataset will be valuable in training a learning model by observing the network packet exchanges and QoE. Such a model can predict the QoE of other video streaming applications that use similar packet exchanges like YouTube.

## 7 Conclusion

In this paper, we presented H3B, a toolbox to collect application and network layer logs for YouTube video streaming. Such a tool can emulate any given network pattern, which is one of its major features. We utilized this tool to launch a measurement campaign for 5 different geographical locations and 5 different bandwidth patterns. We obtained a dataset of 5464 streaming hours over 5936 sessions of YouTube video streaming. We further analyzed the dataset and observed that HTTP/3 is not always a winner compared to legacy HTTP/2. We believe our dataset will be valuable to the community for developing various solutions to provide better application QoE on top of the HTTP/3 browsers.

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
