# —Supplementary Material—
# A Dataset for Analyzing Streaming Media Performance over HTTP/3 Browsers

**Sapna Chaudhary**
IIIT Delhi
sapnac@iiitd.ac.in

**Naval Kumar Shukla**
IIIT Delhi
naval19065@iiitd.ac.in

**Sandip Chakraborty**
IIT Kharagpur
sandipc@cse.iitkgp.ac.in

**Mukulika Maity**
IIIT Delhi
mukulika@iiitd.ac.in

## 1 Methods to generate mahimahi packet delivery trace file:

In this paper, we have generated the mahimahi packet delivery trace file for two cases:

1. For emulating certain bandwidth patterns.

2. For emulating a pcaps collected in real-time.

### 1.1 Generation of mahimahi packet delivery trace file for different bandwidth patterns:

We have emulated Dynamic High (DH), Dynamic Low (DL), and Dynamic Very Low (DVL) bandwidth patterns. Each line in the trace file represents the time at which the packet of size MTU (Maximum Transmission Unit) can be delivered. This transmission time is decided based on the bandwidth during that time instant. For example, to create a 64-256-64-inc (DVL) bandwidth pattern trace file, where the starting bandwidth is 64kbps, the last bandwidth is 256kbps, and the jump required is 64kbps after every 60 seconds. We follow the steps as given below:

- The first packet goes at time t=0.

- For the second packet, say the bandwidth is 64000bps, then the next packet of size MTU(1500 bytes) transmission time will be (0+(1500*8)/64000) = 0.18 second.

- Say after 60 seconds the bandwidth is 128kbps, and the last packet was send at time $t_i$ then next packet transmission time will be $(t_i+(1500*8)/128000)$.

This pattern from start to last bandwidth and then back from last to start bandwidth repeats in a cyclic fashion and based on that the trace file is created.

### 1.2 Generation of mahimahi packet delivery trace file from packet capture files:

To emulate packet captures (pcap) collected in a network, we have converted them into packet delivery trace files (supported by Mahimahi). For conversion, we have used a mechanism used in [1]. The steps are as follows:

Submitted to the 37th Conference on Neural Information Processing Systems (NeurIPS 2023) Track on Datasets and Benchmarks. Do not distribute.

- We convert pcaps into CSV files and extract relevant fields such as real-tile and length of a packet.
- The length field is used to estimate the throughput, which will be further used to estimate the transmission time of a packet of size MTU.
- Then, based on the estimated throughput, the packets per millisecond are computed as discussed in the above subsection.

## 2   Extraction of QoE parameters from the application logs:

Following Penseive [1], we compute the QoE as follows.

$$QoE = \text{Average Bitrate} - \text{Average Bitrate Variation} - 4.3 \times \text{Average Stall}$$

Where,

$$\text{Average\_Bitrate} = \frac{\Sigma_{i=1}^{n} duration_i * bitrate_i}{\Sigma_{i=1}^{n} duration_i},$$
$$\text{Average\_Bitrate\_Variation} = \frac{\Sigma_{i=1} |bitrate_i - bitrate_{i-1}|}{\Sigma_{i=1}^{n} duration_i},$$
$$\text{Average\_Stall} = \frac{real\_time - playback\_time - \Sigma_{i=1} duration_i}{\Sigma_{i=1}^{n} duration_i}$$

duration = the total duration for which the average bitrate, average bitrate variation, and average stall are to be computed

We compute these parameters from the "steamingstats" field parameter named 'cmt' of the application log. The cmt parameter tells the data in the form 'real_time:playback_time'. For computation, the raw application logs are converted into a JSON file format. Hence, we compute multiple QoE values (from multiple instances of streaming stat) for each streaming session.

### 2.1   Structure of QoE and Network CSV file:

We collected the application logs for HTTP/3-enabled and HTTP/2-enabled browsers. To compare the performance for each collected log, we have created a QoE CSV file using the above-mentioned formula. The structure of the HTTP/3 QoE CSV and network CSV is shown with one sample file shown in table 1 and 2, respectively. In network.csv, protocol number 6 refers to TCP protocol, and number 17 refers to QUIC protocol.

## 3   Dataset Structure Description:

The structure of the data is shown in figure 1

**The GitHub link**: `https://github.com/NKShukla/H3B`

Also, the raw dataset can be **downloaded** using this link: `https://drive.google.com/drive/folders/1MsywvxEPOHagHO6JAQ9FPTGLHV17t638?usp=sharing`

## 4   Authors Statement

We bear the responsibility for any violation of rights, and we also take the responsibility to maintain the GitHub link, and we will address all the issues that will be raised in our GitHub repository. Our dataset is licensed under *GNU-GPL* license.

Table 1: HTTP/3_QoE.csv

| real_time | qoe | bitrate | avg_bitrate | avg_bitrate_variation | avg_stall |
|---|---|---|---|---|---|
| 0.679 | 14.40442308 | 104630 | 104630 | 0 | 20.98269231 |
| 7.238 | 20.8531 | 104630 | 104630 | 0 | 19.483 |
| 16.939 | 26.43679333 | 104630 | 104630 | 0 | 18.18446667 |
| 16.939 | 31.24485625 | 104630 | 104630 | 0 | 17.0663125 |
| 17.04 | 34.52558235 | 104630 | 104630 | 0 | 16.30335294 |
| 17.702 | 38.39901111 | 104630 | 104630 | 0 | 15.40255556 |
| 22.755 | 41.88507895 | 104630 | 104630 | 0 | 14.59184211 |
| 26.201 | 39.547135 | 104630 | 104630 | 0 | 15.13555 |
| 26.201 | 42.4235381 | 104630 | 104630 | 0 | 14.46661905 |
| 26.201 | 44.55763182 | 104630 | 104630 | 0 | 13.97031818 |
| 26.201 | 47.1685625 | 104630 | 104630 | 0 | 13.363125 |
| 26.995 | 53.02320323 | 104630 | 104630 | 0 | 12.00158065 |
| 27.954 | 54.63631875 | 104630 | 104630 | 0 | 11.6264375 |
| 35.543 | 56.15114848 | 104630 | 104630 | 0 | 11.27415152 |
| 35.543 | 57.27220294 | 104630 | 104630 | 0 | 11.01344118 |
| 35.638 | 57.75901714 | 104630 | 104630 | 0 | 10.90022857 |
| 36.295 | 59.04988056 | 104630 | 104630 | 0 | 10.60002778 |
| 38.399 | 60.28177568 | 104630 | 104630 | 0 | 10.31354054 |
| 44.66 | 61.44872105 | 104630 | 104630 | 0 | 10.04215789 |
| 44.66 | 54.9796 | 104630 | 104630 | 0 | 11.54660465 |
| 44.66 | 52.45643182 | 104630 | 104630 | 2788.068182 | 11.485 |
| 44.66 | 52.92048667 | 227305 | 104630 | 2726.111111 | 11.39148889 |
| 44.66 | 54.03619348 | 227305 | 104630 | 2666.847826 | 11.14580435 |
| 44.66 | 51.4243383 | 227305 | 104630 | 6184.978723 | 10.93504255 |

Table 2: network.csv

| protocol | tcp_bytes | quic_bytes | real_time |
|---|---|---|---|
| 6 | 1430 | | 0.1728 |
| 6 | 104 | | 0.1733 |
| 6 | 1430 | | 1.1739 |
| 6 | 1430 | | 1.1861 |
| 6 | 0 | | 1.1892 |
| 6 | 1430 | | 1.1981 |
| 6 | 1430 | | 1.2117 |
| 17 | | 1354 | 1.2159 |
| 17 | | 257 | 1.216 |
| 17 | | 43 | 1.2161 |
| 17 | | 42 | 1.2176 |
| 17 | | 41 | 1.2177 |
| 17 | | 283 | 1.2178 |
| 6 | 1430 | | 1.2221 |

## References

[1] H. Mao, R. Netravali, and M. Alizadeh. Neural adaptive video streaming with pensieve. In *ACM SIGCOMM*, pages 197–210, 2017.

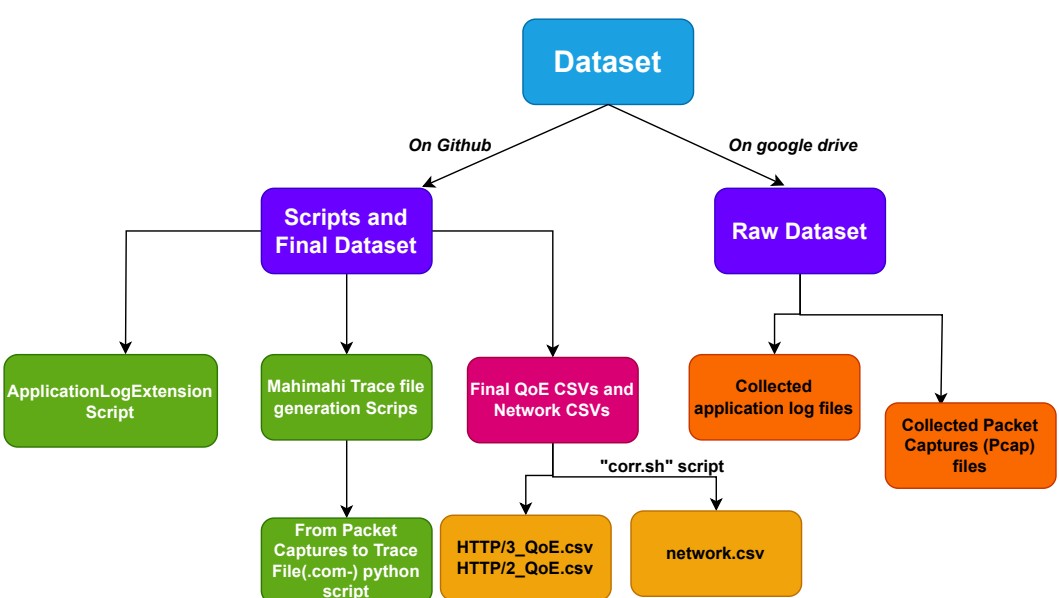

Figure 1: Dataset Structure