# OpenReview forum: "A Dataset for Analyzing Streaming Media Performance over HTTP/3 Browsers"
_NeurIPS.cc/2023/Track/Datasets_and_Benchmarks — NeurIPS 2023 Datasets and Benchmarks Poster_

### Official Review · Reviewer_eSYT · 2023-07-20
**Timely dataset for analyzing HTTP/3**

**Rating:** 7
**Confidence:** 3
**Correctness:** The claims made in the paper are corr…
**Clarity:** Well written.

**Strengths:**

This paper is well-written and targets an important topic: streaming performance under the HTTP3 protocol.  The dataset includes network logs from various network conditions, browsers, and physical addresses, providing comprehensive coverage. The observations from the dataset are also quite intriguing.

**Additional Feedback:**

How is the data collection overhead of H3B, and does it compete with the browser for CPU resources, leading to potential distortion of the collected log performance data?

**Documentation:**

Yes.

**Ethics:**

No.

**Limitations:**

If I have to be picky, by exploring and discussing additional use cases for HTTP/3 beyond media streaming, the paper could offer a more comprehensive perspective on the protocol's versatility and potential benefits in various scenarios. This could enhance the significance and relevance of the research, as it would demonstrate the broader impact of HTTP/3 in diverse real-world applications.

**Opportunities For Improvement:**

Even though video streaming is one of the most important use cases for HTTP/3, it also has other applications, such as file transfers, IoT devices, and more. If the authors can include additional use cases, it will make this article more robust.

**Relation To Prior Work:**

Clearly discussed.

**Summary And Contributions:**

This paper proposes a dataset for analyzing the streaming performance over HTTP/3 browsers named H3B. H3B collects the application and network-level logs during YouTube streaming. Using this tool, we collected a dataset of over 5910 YouTube sessions covering 5445 hours of streaming over 5 different geographical locations and 4 different bandwidth patterns. We believe our tool and as well as the dataset could be used in multiple applications such as a better configuration of application/transport protocols based on the network conditions, intelligent integration of network and application, predicting YouTube's QoE etc. We analyze the dataset and observe that during an HTTP/3 streaming not all requests are served by HTTP/3. Instead whenever the network condition is not favorable the browser chooses to fallback, and the application requests are transmitted using HTTP/2 over the old-standing transport protocol TCP. We observe that such switching of protocols impacts the performance of video streaming applications.

---

> ### Author Response · Authors · 2023-08-11
> **Regarding other usecases of HTTP/3 and data collection overhead**
>
> Thanks for your comments and suggestions. In the future, we will look into other use cases of HTTP/3.
>
> The data collection has less than 25% CPU and 0.6% memory utilization, hence there was no potential distortion to the collected traces.

---

### Official Review · Reviewer_q9Aw · 2023-07-21
**A tool for measuring video streams in a variety of situations and a considerable dataset**

**Rating:** 6
**Confidence:** 3

**Strengths:**

- The H3B tool has been instrumental in collecting network-application data for YouTube streaming sessions. It has efficiently integrated YouTube's "stats for nerds" feature.
- The authors have gathered an impressive dataset from 5445 hours of YouTube streaming sessions across diverse geographical locations and bandwidth conditions.
- The authors discovered the unexpected fallback to TCP during HTTP/3 streaming sessions. This observation has profound implications for understanding QUIC's performance in real-world scenarios, and more importantly, it questions the purported superiority of QUIC.

**Additional Feedback:**

The questions and suggestions are included in the sections above.

**Clarity:**

The paper is well-written in the following three aspects:

- Clarity: The paper explains the technical aspects of their research, like HTTP/3, QUIC, network bandwidth, and the H3B tool in a manner that should be accessible to readers familiar with the field.

- Structure: The paper provides a clear introduction to their work, details their methodology, discusses their findings, and even compares their work with previous research. This is indicative of a well-structured paper.

- Detail: The paper is comprehensive in detailing the experimental setup, data collection, and analysis methods.

**Correctness:**

Based on the information provided, the claims in the paper appear to be correct.

**Documentation:**

The detail is sufficient.

- The authors do detail the methodology of data collection, mentioning the tools used, locations chosen, and parameters tested. The hierarchy of the data is also discussed, giving us an idea of how it is structured.

- A github URL for reviewer access to the dataset is provided.

**Ethics:**

No ethical concerns is raised.

**Limitations:**

- YouTube's stats only provides information about video bitrate and quality levels represented by only resolutions like 720p or 1080p, and some evaluation metrics for visual quality are missing. Due to the concaveness of the quality-bitrate curve, it's hard to measure the visual quality with only bitrate. To better support the research about visual quality, some metrics like PSNR, SSIM, or VMAF should be included.
- The modification of Chromium to disable fallback to TCP, although an interesting experiment, might not reflect real-world scenarios where such a fallback mechanism is essential to ensure continuity of service when QUIC might be blocked or less efficient. Thus, the applicability of these results may be limited.
- The study collected different numbers of sessions for Firefox and Chromium browsers. The unequal number of sessions might influence the analysis and comparison between the two browsers.
- More TCP or QUIC information should be directly recorded in network.csv provided by H3B to better support future works on HTTP/3 video streaming. In the file provided by the authors, only tcp_bytes, quic_bytes and time stamps are provided, but information like packet loss rate is also important, since it is utilized by the state-of-the-art ABR algorithms in different scenarios like [1] and [2] to gain better performance.

[1] Yan F Y, Ayers H, Zhu C, et al. Learning in situ: a randomized experiment in video streaming[C]//17th USENIX Symposium on Networked Systems Design and Implementation (NSDI 20). 2020: 495-511.
[2] Lee I, Kim S, Sathyanarayana S, et al. R-FEC: RL-based FEC Adjustment for Better QoE in WebRTC[C]//Proceedings of the 30th ACM International Conference on Multimedia. 2022: 2948-2956.

**Opportunities For Improvement:**

Besides the authors' claim in the Section Discussion, there are some points for improvement:
- Though QoE is a significant metric, it does not encompass the entire performance picture. Exploring other metrics such as rebuffering time, average bitrate, and packet loss rate could enrich the comparison between HTTP/2 and HTTP/3. This approach would provide a more detailed and nuanced understanding of the performances of the different protocols.
- The study focuses solely on YouTube. Exploring other applications that are sensitive to network conditions, such as online gaming, video conferencing, or other streaming platforms, could broaden the applicability of the findings.
- While the authors have demonstrated that protocol switching occurs and seems to hinder HTTP/3's performance, the study could benefit from a more in-depth analysis of the exact conditions under which switching happens and its direct impact on QoE.

**Relation To Prior Work:**

The authors clearly distinguish their work from prior contributions.

**Summary And Contributions:**

Summary:

The authors of this paper detail the development and utilization of HTTP/3, a novel application layer protocol that employs QUIC, an element adopted by major browsers due to its faster connection and reduced latency. They introduce H3B, a tool that collects YouTube streaming data, using the advantages offered by QUIC. A comprehensive dataset was collected through this tool, encompassing 5910 sessions or 5445 streaming hours across varied geographical locations and bandwidths. The paper suggests this dataset could be instrumental in refining protocols and predicting Quality of Experience (QoE). However, an intriguing discovery was made that during adverse network conditions, the system defaults to HTTP/2 over TCP, a shift that can negatively affect streaming performance.

Contribution:

- The authors provide a comprehensive tool named H3B and its associated large-scale dataset, examining the performance of HTTP/3 and HTTP/2 during YouTube streaming sessions.
- The authors observed that during HTTP/3 streaming, data often reverts to TCP, impacting performance. This contradiction to expected behavior led them to further investigations and modifications of browser source code, observing improvements in QoE when protocol switching was forcibly stopped.
- The proposed dataset showcases not just diverse bandwidth patterns but also geographical locations, protocols, and video genres, collected over an extended period, which can help to emulate various network behaviors.

---

> ### Author Response · Authors · 2023-08-09
> **Regarding other metrics of QoE, when fallback is forcibly stopped, comparison of Chromium and Firefox**
>
> Regarding other metrics: Please note that in our formulation of QoE, we utilize both average bitrate and rebuffering time. It is defined as QoE = Avg. Bitrate − Avg. Bitrate Variation − 4.3 × Avg. Stall (line 187) used by Pensieve [21]. Note that packet loss rate is not an application layer metric as it will be masked by the transport layer to the application layer.
>
> Solely on YouTube: YouTube is one of the most popular streaming platforms, hence we focus on YouTube measurement in this paper. Further, other streaming platforms such as Netflix, Prime Video, etc. do not support HTTP/3. Video conferencing applications typically use WebRTC and do not use QUIC for video data transmission. Hence, we focus on YouTube only.
>
> Exact conditions under which protocol switching happens: We explored the code flow of the Chromium web browser [1,2] and observed that browsers provide backward compatibility to sustain connectivity in cases of middleboxes blocking QUIC. The browser races a QUIC connection with a TCP one (prefers the QUIC one by delaying the TCP one) and whichever finishes first gets used for serving the HTTP request. If a QUIC connection suffers i.e., it faces failures or stream errors such as QUIC_STREAM_CONNECT_ERROR or QUIC_STREAM_REQUEST_INCOMPLETE, the browser chooses to fallback to TCP. Now, the browser does not explicitly check whether the connection failure was due to QUIC blockage on the network path or because of a congested network. Conducting a root cause analysis is outside the scope of this paper.
>
> Regarding other visual quality metrics: Since we have the base video and the playback bitrate, we can generate the video at the target quality. Next, use a frame-by-frame comparison for PSNR or SSIM-based computation. However, as indicated in prior work [3] PSNR-like metric fails to capture user engagement in the wild, so we have resorted to formula-based QoE metric.
>
> Modification to stop fallback: Yes, very correctly pointed out that forcibly stopping fallback won't work where QUIC is blocked. Please note that in this work, we forcibly stopped the fallback to validate our observation that whether indeed protocol switching impacts application QoE. Developing an intelligent fallback algorithm that can distinguish between network congestion and QUIC being blocked is an interesting future work as both of them provide similar signatures.
>
> Chromium vs Firefox: Please note that we are never comparing Chromium with Firefox. We are comparing HTTP/3 streaming over Chromium with legacy HTTP/2 streaming over Chromium only; the same is true for Firefox. As correctly pointed out the number of sessions over both browsers is different and the fallback implementation is also different. Our objective in this paper is to investigate whether our observation of protocol switching impacts application QoE remains true across browsers.
>
> Regarding information in network.csv: Please note that our network.csv is generated using packet capture (pcap). In this paper, we were interested in correlating the amount of TCP and QUIC bytes with application QoE. Hence, we extracted only 5 fields. But a pcap contains many other fields apart from the 5 we reported. We can easily add other relevant information.  Since such a collection is done at the receiver side, we will never receive a lost packet. But, one can determine retransmitted packets and we can that in the network.csv
>
>
> [1] A. Langley, A. Riddoch, A. Wilk, A. Vicente, C. Krasic, D. Zhang, F. Yang, F. Kouranov, I. Swett, J. Iyengar et al., “The quic transport protocol: Design and internet-scale deployment,” in Proceedings of the conference of the ACM special interest group on data communication,
> 2017.
> [2]  R. Hamilton, “Re: [QUIC] graceful fallback to tcp,” https://mailarchive.ietf.org/arch/msg/quic/ph1IAVBa5pW1AgDvr8fbD741Auw/, 2016
> [3] Balachandran, Athula, Vyas Sekar, Aditya Akella, Srinivasan Seshan, Ion Stoica, and Hui Zhang. "Developing a predictive model of quality of experience for internet video." ACM SIGCOMM Computer Communication Review 43, no. 4 (2013): 339-350.

---

> > ### Comment · Reviewer_q9Aw · 2023-08-17
> > **RE: Regarding other metrics of QoE, when fallback is forcibly stopped, comparison of Chromium and Firefox**
> >
> > Thank you for your comments.
> > > Regarding other metrics: Please note that in our formulation of QoE, we utilize both average bitrate and rebuffering time.
> >
> > There is no widely accepted standard for the definition of QoE. Therefore, most solutions still need to evaluate the metrics involved, including rebufferring, average bitrate, etc. Even Pensieve [1] (a paper from a long time ago, without discussing whether it is currently considered a SOTA solution) has done related experiments.
> >
> > > Regarding other visual quality metrics: Since we have the base video and the playback bitrate, we can generate the video at the target quality.
> >
> > The reviewer understands the challenges with metrics like PSNR and SSIM in capturing user engagement. However, the reviewer's point was more about the potential richness added by introducing visual quality metrics. For example, two videos with same average bitrate but different content complexity will definitely have different visual quality and QoE. It's crucial for the future video delivery works to take the content complexity of the videos into consideration, thus the visual quality metrics are indispensable in the proposed dataset. At least, metrics like PSNR and SSIM could be utilized to measure the impact of video content complexity on QoE.
> >
> > > Regarding information in network.csv: Please note that our network.csv is generated using packet capture (pcap).
> >
> > Although the authors state that it's easy for them to record more information, these information is not actually provided in the dataset for other researchers. It could be better to record other relevant metrics, point out all the information that pcap could provide clearly in the paper and make it an option for the researchers to use.
> >
> > Once again, thank you for your efforts and clarifications.
> >
> > [1] Mao, Hongzi, Ravi Netravali, and Mohammad Alizadeh. "Neural adaptive video streaming with pensieve." Proceedings of the conference of the ACM special interest group on data communication. 2017.

---

> > > ### Author Response · Authors · 2023-08-22
> > > **Regarding QoE and other parameters of pcaps**
> > >
> > > Thanks a lot for your comment, and we appreciate your suggestions about extending the metric space in the dataset for QoE analysis.
> > >
> > > We agree that there is no widely adopted definition of QoE; we have used the QoE formula initially proposed in MPC[1] and subsequently adopted by several recent works, including Pensieve [2,3,4]. The formula is designed not to give preference to only one contributing factor, but it's a general QoE formula that is flexible enough to model varying user preferences. Hence, we choose this formulation of QoE.
> > >
> > > We agree that the two videos with the same average bitrate but different content complexity will have different visual quality and QoE, where PSNR and SSIM can be utilized for analysis. However, one important point to note here is that the primary objective of this dataset is to compare the performance of the streaming media QoE over two different network protocols -- QUIC and TCP while keeping the network bandwidth and the video content the same. Such an analysis would be important to design the streaming protocols for HTTP/3 by optimizing the underlying TCP and QUIC connections and designing the best usage for those two protocols. As the video content remains the same, and the underlying network protocol for streaming varies, it would not impact the SSIM and PSNR much.
> > >
> > > For this particular case, as shown in the existing literature [1,2,3,4], average bitrate, bitrate variation, and rebuffering give the best indicator of the QoE. However, as we have stated earlier, PSNR and SSIM can be computed for the analysis from the raw video data and the streaming bitrate information as given in our dataset, which one can utilize as additional analysis metrics.
> > >
> > > In Section 3.1 of the revised paper, we have included the relevant information that would be useful to extract the metrics from the pcap data, as per your suggestion.
> > >
> > >
> > > [1] Yin, X., Jindal, A., Sekar, V. and Sinopoli, B., 2015, August. A control-theoretic approach for dynamic adaptive video streaming over HTTP. In Proceedings of the 2015 ACM Conference on Special Interest Group on Data Communication (pp. 325-338)
> > >
> > > [2] Mao, H., Netravali, R. and Alizadeh, M., 2017, August. Neural adaptive video streaming with Pensieve. In Proceedings of the conference of the ACM special interest group on data communication (pp. 197-210).
> > >
> > > [3] Yan, F.Y., Ayers, H., Zhu, C., Fouladi, S., Hong, J., Zhang, K., Levis, P. and Winstein, K., 2020. Learning in situ: a randomized experiment in video streaming. In 17th USENIX Symposium on Networked Systems Design and Implementation (NSDI 20) (pp. 495-511).
> > >
> > > [4] Meng, J., Xu, Q. and Hu, Y.C., 2021. Proactive {energy-aware} adaptive video streaming on mobile devices. In 2021 USENIX Annual Technical Conference (USENIX ATC 21) (pp. 303-316).

---

### Official Review · Reviewer_WFF1 · 2023-07-21
**Review for H3B paper**

**Rating:** 5
**Confidence:** 3
**Clarity:** The paper is clearly written and stra…

**Strengths:**

* I liked that the authors provided future research topics that this dataset enables.
* It is good that the authors integrated their instrumentation with two different browsers, spanning to different engines.
* I also enjoyed the fact that the authors traced videos throughout time and across different browser versions, although they did not seem to analyse this dimension.

**Additional Feedback:**

Although I enjoyed reading the paper and it has merit, I am not sure how well it fits to the audience of NeurIPS. To me, it seems more of a measurements paper for a systems-related venue.

**Correctness:**

The claims that the authors make wrt quality of experience and HTTP/3 fallback seems to be correct.

**Documentation:**

The authors seem to have enough information about replicating their study. They provide both the tool and the traced dataset.
However, they do not mention maintenance plans.
Last, the organisation of the repository could be substantially improved.

**Limitations:**

Although the authors have successfully provided a dedicated list of limitations, I felt that the following could also be mentioned:

* Variety of available video qualities (e.g. HDR, 4k, high framerate)
* The different browser versions were not really utilised in the provided analysis.
* The clients could be more diverse, spanning across different Operating Systems and platforms.

**Opportunities For Improvement:**

* It would have been insightful if the authors had also integrated a mobile client in their experiments, potentially over cellular connection and analyse the traffic over HTTP/3 there.
* Vantage points were limited and do not really cover severely bandwidth constrained locations.
* The YouTube videos visited are quite long and are on the tail of distribution of the browsing experience, especially if we take into consideration short video uploads such as youtube shorts, instagram reels or tiktok videos. I felt that this was a missed opportunity.

**Relation To Prior Work:**

Related work could be made stronger, as currently the authors very briefly refer to two pieces of work at a very high level and do not highlight how their work substantially differentiates.

**Summary And Contributions:**

This paper publishes a dataset of quality metrics from YouTube streaming over HTTP/3 and HTTP/2 protocols.
By instrumenting two web browsers, namely Chrome and Firefox and controlling the bandwidth and vantage points of traffic, they gather 5910 youtube sessions covevering 5.4k hours of streaming.

Ultimately, the goal of this dataset is to further enable smarter configurations and prediction of QoE over HTTP/3 streaming traffic and the smarter interaction of client application and network traffic.

---

> ### Author Response · Authors · 2023-08-09
> **Regarding mobile client, vantage points, video quality, short videos, suitability to NeurIPS**
>
> Thanks a lot for your detailed and insightful comments. Here are our responses to this.
>
> Regarding available video qualities: In this paper, we focus on video streaming datasets for poor or variable network bandwidth. Telecom Regulatory Authority of India (TRAI) reports that more than 47 million Internet users in India have a download speed of <512 Kbps [1]. We observe that with a bandwidth of 128Kbps, 144p can be achieved and with more than 2Mbps bandwidth 1080p is achieved. Hence, we take videos with maximum quality of 1080p.
>
> Regarding mobile clients: Thank your suggestion. We have collected “in-the-wild” dataset (packet captures) from 10 different users from 4 developing countries: India, Pakistan, Saudi Arabia, and Kenya.  The users carried an Android phone that was connected to the Internet via a cellular connection and were asked to watch YouTube videos. We collected the dataset while they were traveling. We have added this in the github repo (https://github.com/NKShukla/H3B/tree/main/network-log-csv-from-volunteers).  We then replayed these packet captures using our H3B tool and the collected application and network logs. We have included this dataset as well in the github repo and will also add it in the paper (https://github.com/NKShukla/H3B/tree/main/App-Net-volunteers-data)
>
> Vantage points: We have collected data using 5 vantage points across 5 geographical datasets. Along with that we also replay the traces that were collected in a WiFi deployment. Further, we have data from 14 more cities from 4 developing countries.
>
> Short videos: In this paper, our objective is to analyze the performance of video streaming where the Internet bandwidth is poor and intermittent. We believe that for such a case long videos are better suited as this requires long-lived Internet connection. For short videos, the intermediate changes in the network bandwidth do not impact much because the entire video data gets downloaded and buffered within a short duration of time, mostly within the first few seconds of startup.
>
> Browser versions: Since our measurement campaign was launched in 5 different locations over 18 months, our dataset is from  5 different Chrome versions namely 93.0.4577.63, 95.0.4638.54, 95.0.4638.69, 103.0.5025.0, 105.0.0.0 and 1 Firefox version:
> 105.0. We will mention these details in our revised paper.
>
> Client version, platforms: We have conducted measurements on desktops and smartphones. The different OS we have experimented upon are Ubuntu 18.04, 20.04, Android  9, 10, 11, and 12.
>
> Related work: We plan to revise the related work by discussing other related datasets such as the DASH dataset, QoE dataset, and how they are different from ours. Further, we will also discuss the current related work in more depth.
>
> Github repo: We plan to maintain the Github link by regularly pushing updates as we keep on collecting new datasets.  In addition, we will be actively checking the issues that users might be facing.
>
> How it fits to NeurIPS dataset: There are multiple ML/DL-based data-driven ABR (adaptive Bitrate) streaming algorithms and as pointed out by multiple pieces of literature [ 2, 3, 4, 5] such data-driven algorithms fail to perform well in real networks as was seen in an emulation/simulation. To ensure that these ABR streaming algorithms remain robust in poor networks, there is a need for datasets collected under such bandwidth. However, there are limited datasets available for the same, leaving the ML-based ABR streaming algorithms to underperform. Hence, we believe our dataset can aid in designing better ABR algorithms suited to poor bandwidth networks as well.
>
>
> [1]  Telecom Regulatory Authority of India. The Indian Telecom Services Performance Indicators.
> https://www.trai.gov.in/sites/default/files/QPIR_27082021.pdf, 2021
> [2] Yan, Francis Y., Hudson Ayers, Chenzhi Zhu, Sadjad Fouladi, James Hong, Keyi Zhang, Philip Levis, and Keith Winstein. "Learning in situ: a randomized experiment in video streaming." In 17th USENIX Symposium on Networked Systems Design and Implementation (NSDI 20), pp. 495-511. 2020.
> [3] Bartulovic, Mihovil, Junchen Jiang, Sivaraman Balakrishnan, Vyas Sekar, and Bruno Sinopoli. "Biases in data-driven networking, and what to do about them." In Proceedings of the 16th ACM Workshop on Hot Topics in Networks, pp. 192-198. 2017.
> [4] Paul Crews and Hudson Ayers. CS 244 ’18: Recreating and extending Pensieve, 2018. https://reproducingnetworkresearch.wordpress.com/2018/07/16/cs-244-18-recreating-and-extending-pensieve/.
> [5] Yi Sun, Xiaoqi Yin, Junchen Jiang, Vyas Sekar, Fuyuan Lin, Nanshu Wang, Tao Liu, and Bruno Sinopoli. CS2P: Improving video bitrate selection and adaptation with data-driven throughput prediction. In Proceedings of the 2016 Conference of the ACM SIGCOMM, pages
> 272–285, 2016.

---

> > ### Comment · Reviewer_WFF1 · 2023-08-29
> > **Reply to authors**
> >
> > Thank you for the clarifications and additional experiments. I have now increase my score to 5.

---

### Official Review · Reviewer_dwAt · 2023-07-22
**A Useful Dataset for Multimedia Systems Researcher**

**Rating:** 8
**Confidence:** 4
**Correctness:** The collection seems to be sound.
**Clarity:** The paper is reasonable straight forw…

**Strengths:**

The authors meticulously conducted the experiments over a long period of time to collect the data.

The authors have analyzed the data to confirm that HTTP/3 helps QoE.  They carefully analyze the data and found that falling back to TCP hurts the QoE (which is not surprising).

**Additional Feedback:**

Please correct me if I am wrong, but it seems that the raw application log is not included in the dataset (I can't find it on GitHub).  If so, it would be useful to share that as well.

**Documentation:**

The submission lists three intended uses, dynamic tuning of protocols, intelligent network applications, prediction/optimization of YouTube QoE.  But it does not explicitly link how the dataset collected can contribute towards solving these problems.

**Ethics:**

I do not see any ethical issues.

**Limitations:**

The submissions listed diversity in networking conditions, locations, and platforms as a limitation.  This is reasonable, but a more diverse client protocol would also be useful (other streaming services besides YouTube)

**Opportunities For Improvement:**

### Major issues:

The QoE data in the data consists of three fields: average bitrate, average bitrate variation, and average stalls.  The QoE is in turned computed using this three fields.  There are numerous formulation of QoE (see [A]).  Information such as the number of stalls and the duration of stalls are important.  The number of switches between different representation (bitrate) is also important.

The network log data contains "details like the timestamp, source, and destination IP address, protocol used, packet length, and so on."  For a dataset paper it is important to explain what "so on" means.  In the sample data, only the five fields above are listed.

### Minor issues:

1. The related work section is too scarce.  It would be useful to highlight the contributions of this submission in the context of other QUIC datasets (e.g, CESNET-QUIC22).

2. A brief background about video streaming over QUIC would help readers new to this topic (given that this is submitted to NeurIPS and not to networking or systems conference)

### Very Minor issue:

1. Footnote 2 could be better formatted
2. Table 1 is too small.



[A] N. Barman and M. G. Martini, "QoE Modeling for HTTP Adaptive Video Streaming–A Survey and Open Challenges," in IEEE Access, vol. 7, pp. 30831-30859, 2019, doi: 10.1109/ACCESS.2019.2901778.

**Relation To Prior Work:**

No.  Relations to other QUIC, DASH, QoE dataset are not clear enough.

**Summary And Contributions:**

This dataset collects a large number of paired network + QoE traces and over five parts of the world and different network conditions, using HTTP/3 and HTTP/2.

---

> ### Author Response · Authors · 2023-08-11
> **Reagrding QoE metric, related work, diversity**
>
> Thanks for your comments. Here are our responses.
>
> Regarding QoE metric: For QoE computation, we utilize both average bitrate and rebuffering time. It is defined as QoE = Avg. Bitrate − Avg. Bitrate Variation − 4.3 × Avg. Stall (line 187)  [21]. Bitrate variation captures the number of switches between different bitrates and stall captures the duration of rebuffering. Further, since we are computing multiple such moving averages over an entire streaming session, the number of such switchings and the stalls are also considered.
>
> Regarding network.csv: Since, for our analysis, we were interested in correlating the amount of TCP and QUIC bytes with application QoE, we extracted only 5 fields. But a pcap contains many other fields apart, we can extract that too.
>
> Related work and background: We are planning to expand the related work section highlighting the differences from DASH, and other such QUIC and QoE datasets. We will include a brief background of video streaming over QUIC.
>
> The CESNET-QUIC dataset [1] is collected in an ISP backbone network. It can be used for network monitoring and classification of encrypted QUIC network traffic. It includes information like QUIC protocol version, QUIC SNI, timestamp, bytes transmitted, etc. Such a dataset is complementary to our work, as our dataset contains application QoE and network layer traces.
>
> Diff from DASH datasets: (1) Our dataset contains the application logs which are used to extract the QoE metrics such as bitrate, bitrate variation, and stalling which are considered important metrics to determine the QoE of any video streaming application. In addition, it contains network logs for in-depth analysis of application behavior. Whereas the DASH-related dataset [39, 40, 41] contains information like video coding techniques, the bitrate/resolution used, and the duration of segments in which the video is broken down. (2) DASH dataset is complementary to our paper as for our comparison between HTTP/3 and legacy, the DASH algo and the codec used remain the same. (3) We believe our dataset can boost DASH algorithms on occasions of poor network and protocol switching
>
> Difference from QoE datasets: Different QoE datasets [42, 43, 44, 45, 46, 47, 48] are available that contain subjective scores of video or PSNR or SSIM, the adaptive bitrate algorithm used along with the coding details.
> (1) Different QoE datasets either have no stalling [42] or fixed stalling events at fixed durations [43], or fixed stalling patterns [46] (2) Length of a video is at max 300sec, whereas each of our video duration is about 3000sec (3) None of the datasets have network logs. We have time-synchronized application and network logs. (4) All datasets are with only one version of HTTP, we provide datasets of two HTTP protocols and two browsers, different locations (5) The datasets contain QoE that lacks temporal patterns such as [48] provides only one rebuffering duration for the entire video. We plan to include these differences in the related work.
>
> Diversity: YouTube is one of the most popular streaming platforms. Further, other streaming platforms such as Netflix, Prime Video, etc. do not yet support HTTP/3.
>
> Future work: We will include how exactly our dataset can be used to conduct future research presented in the paper.
>
> Raw data: We had given a link to our raw data in the supplementary file (https://drive.google.com/drive/folders/1MsywvxEPOHagHO6JAQ9FPTGLHV17t638?usp=sharing), have included it in the github repo
>
> [39] Taraghi, B., et al. 2022, Multi-codec ultra high definition 8k mpeg-dash dataset. In Proceedings of the 13th ACM Multimedia Systems Conference [40] Zabrovskiy, A., et al., 2018, Multi-codec DASH dataset. In Proceedings of the 9th ACM Multimedia Systems Conference [41] Lederer, S., et al., 2012, Dynamic adaptive streaming over HTTP dataset. In Proceedings of the 3rd multimedia systems conference [42] C. Chen, et al., 2014, ‘‘Modeling the time–varying subjective quality of HTTP video streams with rate adaptations,’’ IEEE Trans. Image Process., [43] Z. Duanmu, et al., 2017, ‘‘A quality-of-experience index for streaming video,’’ IEEE J. Sel. Topics Signal Process. [44] C. G. Bampis, et al., 2017, ‘‘Study of temporal effects on subjective video quality of experience,’’ IEEE Trans. Image Process. [45] N. Eswara et al., 2018, ‘‘A continuous QoE evaluation framework for video streaming over HTTP,’’ IEEE Trans. Circuits Syst. Video Technol. [46] D. Ghadiyaram, et al., 2019, ‘‘A subjective and objective study of stalling events in mobile streaming videos,’’ IEEE Trans. Circuits Syst. Video Technol. [47] Z. Duanmu, et al., 2018, ‘‘Quality-of-experience for adaptive streaming videos: An expectation confirmation theory motivated approach,’’ IEEE Trans. Image Process [48] Bampis, C.G., et al., 2021. Towards perceptually optimized adaptive video streaming-a realistic quality of experience database. IEEE Transactions on Image Processing.

---

> > ### Comment · Reviewer_dwAt · 2023-08-11
> > **RE: Reagrding QoE metric, related work, diversity**
> >
> > Thank you for your comments.
> >
> > > Regarding QoE metric: For QoE computation, we utilize both average bitrate and rebuffering time. It is defined as QoE = Avg. Bitrate − Avg. Bitrate Variation − 4.3 × Avg. Stall (line 187) [21].
> >
> > I understand that you are using the above as the QoE.  My point is that there are many other QoE models that require different metrics and your dataset should not contain only the distilled information for computing this specific QoE metric.  You can make your dataset more useful if you share more information (number of switches, intervals between switches, differences in quality between each switch, number of stalls, duration of each stall, etc) so that whoever wants to use your dataset can either (I) derive Avg Bitrate, Avg Bitrate Variation, and Avg Stall themselves; or (ii) derive other variables used in their QoE metric.
> >
> > > Bitrate variation captures the number of switches between different bitrates and stall captures the duration of rebuffering. Further, since we are computing multiple such moving averages over an entire streaming session, the number of such switchings and the stalls are also considered.
> >
> > There is a difference between "captured" and "considered", and providing raw measurement data.  For instance, instead of sharing the three fields: Avg Bitrate, Avg Bitrate Variation, and Avg Stall, why not just publish the QoE value in the dataset?  The QoE value already "captures" and "considers" the bitrate and stalls.
> >
> > > Regarding network.csv: Since, for our analysis, we were interested in correlating the amount of TCP and QUIC bytes with application QoE, we extracted only 5 fields. But a pcap contains many other fields apart, we can extract that too.
> >
> > Indeed you _can_ extract that information; but it is important to be precise, when writing a scientific paper, on what you extracted.  Saying "like ... and  so on" when you already listed everything you extracted is not ideal.
> >
> > ### Summary
> >
> > When publishing the dataset, some thoughts have to be put into what information is useful to potential users of this dataset.  I am not convinced that sharing only averages of bitrate, bitrate variation, and stall length is useful as it limits the applicability of this dataset.

---

> > > ### Author Response · Authors · 2023-08-11
> > > **Regarding raw application logs**
> > >
> > > Thanks a lot for your comments. Yes, you are right the dataset should contain all the information such as the number of switches, intervals between switches, differences in quality between each switching, number of stalls, duration of each stall, etc to be used by others. All the raw application log files and their corresponding pcap files were uploaded at: https://drive.google.com/drive/u/0/folders/1MsywvxEPOHagHO6JAQ9FPTGLHV17t638 (Link was given into the supplementary file). We will add the raw application logs in the GitHub repo as well, also will modify the paper to clarify these points.
> > >
> > > Our raw application log looks like the following (note we are showing only relevant parameters)
> > > Sample:
> > > {"documentId":"2A5503D46340C4251DABD099FC9236E4"</b>,"documentLifecycle":"active","frameId":0,"frameType":"outermost_frame","initiator":"https://www.youtube.com","method":"POST","parentFrameId":-1,"requestId":"50","tabId":988103974,"timeStamp":1658308399519.4841,"type":"xmlhttprequest","&cmt=0.184:0.000,2.101:0.000}
> > > {"documentId":"2A5503D46340C4251DABD099FC9236E4","documentLifecycle":"active","frameId":0,"frameType":"outermost_frame","initiator":"https://www.youtube.com","method":"GET","parentFrameId":-1,"requestId":"51","tabId":988103974,"timeStamp":1658308399547.0632,"type":"xmlhttprequest",&itag=397}
> > >
> > > We convert the raw application log to json format:
> > >
> > > "50": {
> > >         "type": "streamingstats",
> > >         "request_ts": 0.701034912109375,
> > >         "itag": 397,
> > >         "buffer_health": "2.101:0.00",
> > >         "cmt": "0.184:0.000,2.101:0.000",   // At time t=0.184 sec the initial buffering was 0.184-0.000 = 0.184 and time t=2.101sec the initial rebuffering was 2.101-0.000 = 2.101 sec; Hence, at time t=2.101(real time) the video was still buffering(initial buffering) i.e the the playback time was 0.000;
> > >         "bwe": "2.101:130000",
> > >         "vps": "0.000:N,0.184:B,2.101:B,2.101:B"
> > >     },
> > >     "51": {
> > >         "type": "videoplayback",
> > >         "request_ts": 0.162063232421875,
> > >         "complete_ts": 6.318923095703125,
> > >         "total_time": 6.15685986328125,
> > >         "total_bytes": 174508,
> > >         "complete_range": [
> > >             0,
> > >             174508
> > >         ],
> > >         "complete_itag": 397,   // Mapped with the bitrate given in video info file
> > >         "complete_rbuf_sec": 0.0,
> > >         "complete_rbuf": 0,
> > >         "complete_rn": 1,
> > >         "complete_clen": 150413396,
> > >         "complete_dur": 2624.64,
> > >         "kbytes/second": 27679.36456802463
> > >     },
> > >
> > > Our paper shows the JSON format of application logs in the paper Sec 3.3., page 5 and Table 3.
> > > We have added one sample raw application log file and its corresponding json file here for reference https://drive.google.com/drive/u/0/folders/1jgqLYRluBXygluNsF69niuGtaJXYgoMp
> > >
> > > "There is a difference between "captured" and "considered", and providing raw measurement data. For instance, instead of sharing the three fields: Avg Bitrate, Avg Bitrate Variation, and Avg Stall, why not just publish the QoE value in the dataset? The QoE value already "captures" and "considers" the bitrate and stalls."
> > >
> > > Yes, we agree. Since we have the raw log files all this information can be extracted easily. As suggested we can ourself compute additional information such as the number of switches, intervals between switches, differences in quality between each switch, number of stalls, duration of each stall, etc.
> > > For example, the number of switches can be directly computed by counting the number of times the itag value changes in the application log, interval between the switches can be computed by subtracting the timestamps of the currently changed itag from the previous itag timestamp value, differences in quality between each switch can be obtained by subtracting the bitrates of each itag value. Similarly, the number of stalls and duration of each stall can be computed from the cmt parameter.
> > >
> > >
> > > "Indeed you can extract that information; but it is important to be precise, when writing a scientific paper, on what you extracted. Saying "like ... and so on" when you already listed everything you extracted is not ideal."
> > >
> > > Thanks for your suggestion, we will modify the paper accordingly.

---

> > > > ### Comment · Reviewer_dwAt · 2023-08-29
> > > > **RE: raw application logs**
> > > >
> > > > Adding raw application logs into the dataset addresses my concern.  Thank you.

---

> > > > > ### Author Response · Authors · 2023-08-30
> > > > > **Raw Logs Uploaded on GitHub Repo too**
> > > > >
> > > > > Thanks a lot for your comment. Please note that we have uploaded the raw logs into our Github repo as well (https://github.com/NKShukla/H3B/tree/main/application_layer_raw_logs) in addition to the drive link we had earlier.

---

### Official Review · Reviewer_5bNA · 2023-07-23
**A Dataset for Analyzing Streaming Media Performance over HTTP/3 Browsers**

**Rating:** 3
**Confidence:** 5
**Correctness:** No, the construction of video content…

**Strengths:**

This is the first large-scale YouTube streaming dataset over an HTTP/3 browser. It launches a measurement campaign for YouTube across 5 different geographical locations and 4 different bandwidth patterns and collects 5445 hours of data across 5910 streaming sessions.

**Additional Feedback:**

No

**Clarity:**

No, many details are missing. Many paragraphs can be deleted directly. As Page 4, Mahimahi's explanation can be deleted directly.

**Documentation:**

Yes

**Limitations:**

1. In Subsection 3.1, about the setting "the minimum and maximum video quality of all the videos were 144p and 1080p", the collection of this dataset has little value for future research.
2. Many scholars have done related work before, such as DASH.js and its variants.
3. How do you define the different bandwidth patterns, i.e., high, low, very low, and under-mobility? Why [896Kbps, 1152Kbps] is DH (a good bandwidth) bandwidth pattern? Why don't you take the real network bandwidth traces (such as 3G/HSDPA,4G/LTE, FCC) and build them by yourself?
4. Where did you cite Table 2?
5. The CSV file about the network log only has five types of information. Maybe you can use the CICFlowMeter to select the characteristics of Pcap packets.


**Opportunities For Improvement:**

This paper's contribution is not significant. The video quality and network bandwidth used in this dataset have little value for future research.

**Relation To Prior Work:**

No, many scholars have done related work before, such as DASH.js and its variants. This paper does not compare with those.

**Summary And Contributions:**

This paper presents an HTTP/3-supported browser dataset collection tool named H3B. Using this tool, this paper collected a dataset of over 5910 YouTube sessions covering 5445 hours of streaming over 5 different geographical locations and 4 different bandwidth patterns.

---

> ### Author Response · Authors · 2023-08-08
> **Regarding bandwidth patterns, video quality and difference from DASH, QoE datasets**
>
> Thanks for your detailed comments. Here are our responses.
>
> Bandwidth patterns: Parts of developing countries like India still lack access to consistent high speed Internet [1, 5, 20]. Though there are datasets of video streaming [11,33], such a collection was not performed specifically for poor or variable network bandwidths. Hence, we used different bandwidth patterns focusing on poor networks and replay network traces that were collected during mobility ina WiFi deployment(Real in Table 4).  [896, 1152] Kbps is considered as good as 720p can be achieved with this.
> Further, we have also performed “in-the-wild” trace collection  for 10 users on their Android smartphones while connected via a cellular network. Such a collection is done while they are traveling and watching YouTube videos. We have included this in github repo (https://github.com/NKShukla/H3B/tree/main/network-log-csv-from-volunteers). We then replayed these traces using our H3B tool and the collected application and network logs. We have included this dataset as well in the github repo and will add in the paper (https://github.com/NKShukla/H3B/tree/main/App-Net-volunteers-data)
>
> Regarding video quality: The Telecom Regulatory Authority of India (TRAI) reports that more than 47 million Internet users in India have a download speed of <512 Kbps [38]. We observe that 144p and 1080p can be achieved with 128Kbps and >2Mbps bandwidth respectively. Hence, we take videos with maximum quality of 1080p ( YouTube announces 1080p as HD)
>
> Diff from DASH datasets:
> (1) Our dataset contains the application logs which are used to extract the QoE metrics such as bitrate, bitrate variation, and stalling which are considered important metrics to determine the QoE of any video streaming application. In addition, it contains network logs for in-depth analysis of application behavior. Whereas the DASH-related dataset [39, 40, 41] contains information like video coding techniques, the bitrate/resolution used, and the duration of segments in which the video is broken down.
> (2) DASH dataset is complementary to our paper as for our comparison between HTTP/3 and legacy, the DASH algo and the codec used remain the same.
> (3) Finally, We believe our dataset can boost DASH algorithms on occasions of poor network and protocol switching
>
> Difference from QoE dataset: Different QoE datasets [42, 43, 44, 45, 46, 47, 48] are available that contain subjective scores of video or PSNR or SSIM, the adaptive bitrate algorithm used along with the coding details.
> (1) Different QoE datasets either have no stalling [42] or fixed stalling events at fixed durations [43], or fixed stalling patterns [46]
> (2) Length of a video is at max 300sec, whereas each of our video duration is about 3000sec
> (3) None of the datasets have network logs in addition to application logs. We have time-synchronized application and network logs  that can be used to better characterize the impact of the network on application QoE
> (4) All datasets are of HAS (HTTP Adaptive Streaming) with only one version of HTTP,  we provide with two HTTP protocols and two web browsers, different locations
> (5) The datasets contain QoE information in an aggregated fashion  that lacks temporal patterns such as [48] provides only one rebuffering duration for the entire video
>
> We plan to include these differences in the related work.
>
> [38] Telecom Regulatory Authority of India. The Indian Telecom Services Performance Indicators.
> https://www.trai.gov.in/sites/default/files/QPIR_27082021.pdf, 2021
> [39] Taraghi, B., et al. 2022, Multi-codec ultra high definition 8k mpeg-dash dataset. In Proceedings of the 13th ACM Multimedia Systems Conference
> [40] Zabrovskiy, A., et al., 2018, Multi-codec DASH dataset. In Proceedings of the 9th ACM Multimedia Systems Conference
> [41] Lederer, S., et al., 2012,  Dynamic adaptive streaming over HTTP dataset. In Proceedings of the 3rd multimedia systems conference
> [42] C. Chen, et al., 2014, ‘‘Modeling the time–varying subjective quality of HTTP video streams with rate adaptations,’’ IEEE Trans. Image Process.,
> [43] Z. Duanmu, et al., 2017, ‘‘A quality-of-experience index for streaming video,’’ IEEE J. Sel. Topics Signal Process.
> [44] C. G. Bampis, et al., 2017, ‘‘Study of temporal effects on subjective video quality of experience,’’ IEEE Trans. Image Process.
> [45] N. Eswara et al., 2018, ‘‘A continuous QoE evaluation framework for video streaming over HTTP,’’ IEEE Trans. Circuits Syst. Video Technol.
> [46] D. Ghadiyaram, et al., 2019, ‘‘A subjective and objective study of stalling events in mobile streaming videos,’’ IEEE Trans. Circuits Syst. Video Technol.
> [47] Z. Duanmu, et al., 2018, ‘‘Quality-of-experience for adaptive streaming videos: An expectation confirmation theory motivated approach,’’ IEEE Trans. Image Process
> [48] Bampis, C.G., et al., 2021. Towards perceptually optimized adaptive video streaming-a realistic quality of experience database. IEEE Transactions on Image Processing.

---

### Author Response · Authors · 2023-08-11
**Regarding Network setup, Video quality, Related work, Mobile client, Vantage points and QoE metric**

# Video Quality & Network
Parts of developing countries like India still lack access to consistent high speed Internet [1, 5, 20]. 47 million Internet users in India have a download speed of <512 Kbps [38]. Although video streaming datasets exist [11, 33], they often lack a specific focus on poor or fluctuating network conditions. To address this gap, we created diverse bandwidth patterns, concentrating on challenging network bandwidth. Additionally, we replayed network traces collected in a WiFi deployment during mobility.
We take 1080p as max quality as 144p and 1080p can be achieved with 128Kbps and >2Mbps bandwidth respectively.

# Related Work
We plan to expand the related work section to emphasize our distinctions from DASH and other QoE datasets.
Diff from DASH datasets: (1) Our dataset includes application logs for QoE metrics like bitrate, variation, and stalling, along with network logs for detailed application analysis. In contrast, DASH datasets [39, 40, 41] mainly cover video coding techniques, bitrates, and segment durations. (2) DASH datasets complement ours as both HTTP/3 and legacy use the same algorithm and codec, (3)
Our dataset can enhance DASH algorithms for poor network scenarios and protocol switching.

# Diff from QoE Datasets
[42, 43, 44, 45, 46, 47, 48]
(1) Existing QoE datasets lack stalling [42], have fixed stalling events [43], or patterns [46].
(2) Our video durations are up to 3600s, compared to a maximum of 300s in other datasets.
(3) We uniquely provide time-synced application and network logs to better understand network impact on QoE.
(4) Our dataset encompasses two HTTP protocols, two browsers, and diverse locations, unlike single-version QoE datasets.
(5) Existing datasets aggregate QoE data, lacking temporal patterns such as [48] providing one rebuffering duration for an entire video

# Mobile Client Cellular Network
We collected "in-the-wild" packet captures from 10 Android smartphone users watching YouTube videos while traveling via cellular networks, uploaded on our repo (https://github.com/NKShukla/H3B/tree/main/network-log-csv-from-volunteers). We replayed them using our H3B tool, and the dataset is included in the repo and will add in the paper (https://github.com/NKShukla/H3B/tree/main/App-Net-volunteers-data).

# Diff OS, Vantage Points
Our measurements encompass desktops and smartphones (including 5 Chrome and 1 Firefox version). OS: Ubuntu 18.04, 20.04, Android 8, 9, 10, 11, 12.  5 geographical vantage points, replayed trace from WiFi. Additionally, 14 more cities in 4 developing countries.

# QoE Metric
Since we have the base video and the playback bitrate, we can generate the video at the target quality. Next, we can use a frame-by-frame comparison for PSNR or SSIM based computation. However, as indicated in prior work [49] PSNR-like metric fails to capture user engagement in the wild. Furthermore, rebuffering and quality switches are not considered in PSNR and SSIM calculations.

# Other platforms than YouTube
 YouTube is one of the most popular streaming platforms. Other streaming platforms such as Netflix, Prime Video, etc. do not yet support HTTP/3. Video conferencing applications mostly use WebRTC and do not use QUIC.

[38] Telecom Regulatory Authority of India. The Indian Telecom Services Performance Indicators, 2021 https://www.trai.gov.in/sites/default/files/QPIR_27082021.pdf [39] B. Taraghi, et al. 2022, Multi-codec ultra high definition 8k mpeg-dash dataset. In Proceedings of the 13th ACM Multimedia Systems Conference [40] A. Zabrovskiy, et al., 2018, Multi-codec DASH dataset. In Proceedings of the 9th ACM Multimedia Systems Conference [41] S. Lederer, et al., 2012, Dynamic adaptive streaming over HTTP dataset. In Proceedings of the 3rd multimedia systems conference [42] C. Chen, et al., 2014, ‘‘Modeling the time–varying subjective quality of HTTP video streams with rate adaptations,’’ IEEE Trans. Image Process., [43] Z. Duanmu, et al., 2017, ‘‘A quality-of-experience index for streaming video,’’ IEEE J. Sel. Topics Signal Process. [44] C. G. Bampis, et al., 2017, ‘‘Study of temporal effects on subjective video quality of experience,’’ IEEE Trans. Image Process. [45] N. Eswara et al., 2018, ‘‘A continuous QoE evaluation framework for video streaming over HTTP,’’ IEEE Trans. Circuits Syst. Video Technol. [46] D. Ghadiyaram, et al., 2019, ‘‘A subjective and objective study of stalling events in mobile streaming videos,’’ IEEE Trans. Circuits Syst. Video Technol. [47] Z. Duanmu, et al., 2018, ‘‘Quality-of-experience for adaptive streaming videos: An expectation confirmation theory motivated approach,’’ IEEE Trans. Image Process [48] C.G. Bampis, et al., 2021. Towards perceptually optimized adaptive video streaming-a realistic quality of experience database. IEEE Transactions on Image Processing. [49] A. Balachandran, et al. 2013,  "Developing a predictive model of quality of experience for internet video." ACM SIGCOMM Computer Communication Review

---

### Author Response · Authors · 2023-08-21
**Revised manuscript**

Dear reviewers,
We thank you for your detailed and insightful comments. We have revised the paper according to your suggestions. The changes are highlighted in blue.

---

### Decision · Program_Chairs · 2023-09-22

**Decision:**

Accept (Poster)

**Comment:**

The reviewers found this paper interesting and timely, and are overall positive in their evaluation. We recommended accept, but encourage the authors to integrate all feedbacks in the final version, especially those from reviewer 5bNA.